# Care-seeking for diarrheal illness: A systematic review and meta-analysis

Kirsten E. Wiens [1]*, Marissa H. Miller[1], Daniel J. Costello[1,2], Ashlynn P. Solomon[1], Skye M. Hilbert[1], Andrea G. Shipper[3], Elizabeth C. Lee[4], Andrew S. Azman[4,5]

1 Department of Epidemiology and Biostatistics, College of Public Health, Temple University, Philadelphia, United States of America, 2 Department of Epidemiology, Milken Institute School of Public Health, The George Washington University, Washington, District of Columbia, United States of America, 3 Charles Library, Temple University, Philadelphia, United States of America, 4 Department of Epidemiology, Johns Hopkins Bloomberg School of Public Health, Johns Hopkins University, Baltimore, United States of America, 5 Geneva Centre for Emerging Viral Diseases and Division of Tropical and Humanitarian Medicine, Geneva University Hospitals, Geneva, Switzerland

* kirsten.wiens@temple.edu

## Abstract

Monitoring and treating diarrheal illness often relies on individuals seeking care at hospitals or clinics. Cases that seek care through pharmacies and community health workers (CHW) are frequently excluded from disease burden estimates, which are used to allocate mitigation resources. Studies on care-seeking behavior can help identify these gaps but typically focus on children under five, even though diarrheal diseases like cholera and Enterotoxigenic *E. coli* affect all age groups. This study aimed to estimate the proportion of individuals seeking care for themselves or their children with diarrhea, considering different age groups, case definitions, and study settings. We conducted a systematic review of population-based primary research studies published during 2000–2024 that examined care-seeking behavior for diarrhea. We included 166 studies from 62 countries. Hospitals and clinics were the most common source of care sought outside the home, with CHW and health posts rarely reported. Using a random-effects meta-analysis, we found substantial heterogeneity in care-seeking between studies from low- and middle-income countries ($I^2$ = 99.3) and estimated that the proportion of diarrhea cases seeking care at a hospital or clinic was 32.8% on average (95% Confidence Interval (CI) 28.1% to 37.9%; prediction interval 3.3% to 87.5%). Although there were trends toward higher care-seeking for children compared to adults, substantial variation existed between studies, and the differences were not significant. We estimated that the adjusted odds of seeking care at a hospital or clinic were significantly higher for severe diarrhea and cholera compared to general diarrhea (Odds Ratio 3.43; 95% CI 1.71 to 6.88). Our findings confirm that passive surveillance through hospitals and clinics may substantially undercount the number of people with diarrhea, particularly those with milder symptoms, although this proportion varied widely. Additionally, our findings underscore the importance of including care-seeking questions across all age groups in future studies, as we cannot assume lower care-seeking for adults across all settings. Our study was limited by fewer data on care-seeking from health posts, traditional healers, and CHW compared to hospitals and clinics, highlighting a need for further research on these sources of care.

**Data availability statement:** All extracted data and source code for data cleaning and meta-analysis are available at https://github.com/wienslab/diarrhea-careseeking.

**Funding:** This study was supported by the Bill and Melinda Gates Foundation (INV-044865 to ASA) and the National Institute of Allergy and Infectious Diseases (K22AI168389 to KEW). The funders had no role in study design, data collection and analysis, decision to publish, or preparation of the manuscript.

**Competing interests:** The authors have declared that no competing interests exist.

## Introduction

Diarrheal diseases are a leading cause of illness and death worldwide, particularly among children under five [1,2]. Diarrhea burden estimates rely partly on data from population-based surveys focused on children, which use caregiver recall of recent symptoms [1,2]. In adults and for specific pathogens, estimates are often based on passive surveillance data, such as hospital records and vital statistics [1,2]. These data are especially valuable for tracking seasonal outbreaks of pathogens like cholera that may not appear in surveys but depend on individuals seeking care for often severe symptoms [3,4]. A key challenge in accurately assessing diarrheal disease burden by age and etiology is determining how many cases go unreported due to barriers in care-seeking and identifying which populations or illnesses are most likely to be undercounted.

Estimates of diarrhea burden strongly influence investments in disease mitigation resources, including broad investments in improved water and sanitation and more pathogen-specific tools like vaccines. However, diarrheal pathogens vary in the symptoms they cause as well as the age groups they primarily affect. For example, rotavirus, *Escherichia coli*, and *Salmonella* can cause gastroenteritis with diarrhea and/or vomiting, while cholera is marked by acute watery diarrhea and/or vomiting. Rotavirus primarily affects children, whereas cholera is a leading cause of diarrhea illness and deaths across all age groups [1]. Strategies for controlling rotavirus differ from those used for cholera, as its primary mode of transmission is person-to-person via the fecal-oral route, rather than through consuming contaminated food and water. Enteric pathogens also cause a range of disease severity, from mild infection to severe dehydration requiring rehydration to prevent severe outcomes including death. Given these variations, relying solely on population-level estimates of general diarrhea in children and passive surveillance data may not optimize burden reduction efforts.

Programs such as the Demographic and Healthcare Surveys (https://dhsprogram.com/) that assess care-seeking behavior for diarrhea around the world provide an opportunity to examine these biases in disease burden estimates [5–9]. Previous analyses of these data have estimated that the proportion of caregivers that seek care at health facilities for a child with diarrhea was on average 45% in sub-Saharan Africa [10] and 49–51% in West and Central Africa [11], with substantial variation by country [12] and type of care sought [13]. Importantly, these studies all focused on care-seeking for children, and did not incorporate data from studies that have reported care-seeking across age groups including adults [14,15]. In addition, while one of these reviews found that more people sought care for general childhood illnesses when illness was more severe [13], none of these studies examined care-seeking behavior by severity of diarrheal illness or etiology, limiting the applicability of their results.

In this study, we sought to address these knowledge gaps through a systematic review and meta-analysis of surveys conducted during 2000–2024 on diarrhea care-seeking behavior across all age groups and diarrhea case definitions. Our primary research question was: what proportion of individuals of any age seek care at hospitals, clinics, or other sources when they have diarrhea? In addition, we asked: how does seeking care at hospitals or clinics vary by age group, severity of diarrheal illness, and other factors?

## Materials and methods

### Systematic review

We developed the methods for this study based on the Preferred Reporting Items for Systematic Reviews and Meta-Analyses (PRISMA) guidelines [16] and submitted the protocol to the International Prospective Register of Systematic Reviews (PROSPERO) on February 24th, 2023 (review ID: CRD42023402435) [17].

We searched PubMed, Embase, Web of Science, and Global Index Medicus on January 27, 2023 using search terms developed by a medical librarian (AGS); searches were updated on September 3, 2024 (Section A in S1 Appendix). We included studies that 1) included sampling on or after January 1, 2000, 2) were primary research studies, 3) were conducted in human populations, 4) included questions about diarrhea care-seeking, perceptions knowledge, and/or management, 5) surveyed individuals of any age about their own care-seeking for diarrheal illness, 6) reported the number of individuals surveyed about care-seeking for diarrheal illness either for themselves or a child, 7) reported the number or proportion who said that they would or did seek care, and 8) were written in English, French or Spanish. We excluded studies that 1) were conducted in special populations, e.g., HIV- or malaria-infected individuals, healthcare professionals, deployed troops, travelers, or long-term care facility residents, 2) focused exclusively on non-infectious diarrhea, 3) did not separate diarrhea-related care-seeking from other illnesses, 4) did not report the sampling method and study population, 5) were case studies, case-control studies where cases selected based on care-seeking behavior, or qualitative studies, 6) used convenience sampling, 7) were reviews or secondary data analyses, 8) were conference abstracts, or 9) did not have full texts available.

Studies were screened by title and abstract by two blinded and independent reviewers (DJC, APS, MHM, or KEW) using Covidence (https://www.covidence.org/). If a tiebreaker was needed, a third reviewer was used, or a decision was made by consensus. This process was repeated for full text screening. For the updated search, a single reviewer (MHM) performed the screening. One of the reviewers independently extracted data from each included study using a shared Google Sheet. Extracted data included study design, sampling method, study population, diarrhea case definition, start and end dates, study catchment area matched to a GADM polygon (https://gadm.org/) when possible, how the care-seeking questions were phrased, who the care was sought for (e.g., self or child), recall time in days, type of care facility sought, number of individuals surveyed about diarrheal illness, whether they "would" and/or "did" seek care, and when available, age and sex distribution of the surveyed population, diarrhea symptoms, and diarrhea severity (S1 Data). When care-seeking data were stratified by any of these variables, we extracted the data at those stratifications. If a study reported "yes", "no", and "maybe" for "would/will you seek care", we coded "maybe" as "no".

A single reviewer (SMH) evaluated the quality of the studies that met our inclusion criteria using an adapted version of the appraisal tool for cross-sectional studies (AXIS) [18]. Specifically, we assessed whether 1) the data presented in the study were internally consistent (i.e., numbers for the proportion seeking care were identical throughout the abstract, text, and tables), 2) the manuscript included a justification for their sample size, and 3) the response rate was reported [0 = no, 1 = yes]. An overall quality score was created by summing across the three indicators for total scores ranging from 0-3, where 0 was lowest quality and 3 was highest quality. Studies were not excluded based on quality assessment criteria. We retained studies with internal inconsistencies to avoid penalizing those that provided more detailed information or in-depth discussions of results, where minor differences or ambiguities often arose. Studies with major ambiguities were excluded in full text screening for lacking care-seeking data (for example, if it was unclear what the denominator was for the reported proportion seeking care).

## Data cleaning

We manually reviewed studies by country, sorted by start date, to identify studies with overlapping data (i.e., conducted among the same people in the same time period with the same questions). Overlapping studies were excluded from the primary dataset, and reason for

selection of the retained study was recorded in the extraction sheet. We prioritized inclusion of studies with 1) more representative sampling methods, 2) more consistent or clearly described data, 3) more specific case definition, 4) larger sample size, 5) longer sampling period, 6) more detailed age information, and 7) more detailed methods description, in that order.

Data were cleaned, checked for mistakes, and analyzed using R statistical software version 4.3.3 [19]. Extracted data were checked for correct location information (e.g., could be correctly matched to a GADM shapefile), impossible values (e.g., number seeking care greater than number surveyed, start date after end date), and inconsistent values (e.g., multiple different sample sizes entered per study or stratification). Data were matched with World Bank 2023–2024 Income Classifications [20] and regions by country.

We created three standardized categories for diarrhea case definition: 1) "diarrhea" included explicit definitions of three or more loose stools in past 24 hours and diarrhea used broadly without an explicit definition; 2) "severe diarrhea or cholera" included severe diarrhea, defined as having danger signs, dehydration, ≥1 week duration, or death, and acute watery diarrhea of any severity including deaths; 3) "gastroenteritis or other etiologies" included definitions of either diarrhea or vomiting, and specific etiologies including rotavirus, *Escherichia coli*, *Salmonella*, and *Giardia*. A single study stratified care-seeking by diarrhea cases that were "moderate to severe" (diarrhea with sunken eyes, loss of skin turgor, dysentery, intravenous rehydration, or hospitalization) and "less severe" (all other diarrhea) [21]; we categorized "moderate to severe" as "severe diarrhea or cholera" and "less severe" as "diarrhea".

In addition, we created standardized categories of the source of care sought (Table A in S1 Appendix). For the analysis of care-seeking at hospitals or clinics, we defined "hospital or clinic" as any general hospital or clinic or explicitly defined public hospitals or clinics. We excluded explicitly defined private hospitals or clinics from this category because they are rarely part of surveillance systems [22,23].

## Meta-analysis

Observations, or units for analysis, were created by aggregating extracted data by study, country, source of care sought, and potential sources of variation in care-seeking (shown in Table 1). For studies with multiple-choice questions that provided several options within the same category (e.g., multiple different public hospitals and/or community clinics under 'source of care'), we used the highest percentage reported within the category to represent the overall care-seeking behavior for that category in our analysis.

We used logistic regression models with observation-level random intercepts to pool estimates of the proportion of individuals with diarrhea that did or would seek care for diarrhea at hospitals or clinics across all included studies, implemented using the metaprop function of the meta package in R [24]. We used logistic regression models with observation-level random intercepts and fixed effects to examine the effects of diarrhea case definition, whether the study was in an urban or rural location, whether the study took place during or closely following a diarrhea outbreak, as well as other components of study methodology on care-seeking in univariate and multivariate analyses, implemented using the rma function of the metafor package in R [25]. For multivariate analysis, we included variables that we identified as significantly associated with care-seeking (p < 0.05) in univariate analysis but that were not significantly associated with other included variables. In addition, we performed sub-group analyses by age group and diarrhea case definition.

## Data and code availability

All extracted data and source code for data cleaning and meta-analysis are available at https://github.com/wienslab/diarrhea-careseeking.

**Table 1. Study characteristics. Number of observations in the primary dataset with each characteristic. A study has more than one observation if it reported results stratified by any of these characteristics.**

| Category | Characteristic | Number (N = 211) | Percent (%) |
|---|---|---|---|
| Timing of care-seeking | Care sought at any time | 175 | 82.9 |
| | First source sought | 29 | 13.7 |
| | Care sought prior to current hospital visit | 6 | 2.8 |
| | Care sought within 24 hours of symptom onset | 1 | 0.5 |
| Relationship of diarrhea case to survey respondent | Child (of survey respondent) | 138 | 65.4 |
| | Self (survey respondent) | 14 | 6.6 |
| | Combination (child, self, or other family member) | 59 | 28.0 |
| Survey recall period | 2-7 days before interview | 13 | 6.2 |
| | 14-30 days before interview | 143 | 67.8 |
| | 42-365 days before interview | 18 | 8.5 |
| | Not reported | 37 | 17.5 |
| Clinic-based surveillance | No | 198 | 93.8 |
| | Yes | 13 | 6.2 |
| During or recently after an outbreak that causes diarrhea | No | 194 | 91.9 |
| | Yes | 17 | 8.1 |
| Multiple choices for types of care sought | No | 180 | 85.3 |
| | Yes | 31 | 14.7 |
| Study location type | Rural | 83 | 39.3 |
| | Urban | 68 | 32.2 |
| | Urban and Rural | 52 | 24.6 |
| | Peri-Urban | 5 | 2.4 |
| | Internally Displaced Persons or Refugee Camp | 2 | 0.9 |
| | Urban and Peri-Urban | 1 | 0.5 |
| Country World Bank 2023–2024 income group [20] | Lower middle income | 97 | 46.0 |
| | Upper middle income | 44 | 20.9 |
| | Low income | 43 | 20.4 |
| | High income | 27 | 12.8 |
| Diarrhea case definition | Diarrhea | 156 | 73.9 |
| | Gastroenteritis or other etiologies | 33 | 15.6 |
| | Severe diarrhea or cholera | 22 | 10.4 |

## Results

### Study characteristics

We retrieved 7069 studies from four databases, screened 5188 for relevance following de-duplication, assessed 722 full texts for eligibility (S2 Data), and included 188 studies (Fig 1). Of these, 166 studies were non-overlapping and asked participants whether they did seek care for a past diarrheal illness. For the primary analysis, we stratified these study-level data by characteristics shown in Table 1, for a total of 211 observations pertaining to care-seeking outside of participants' homes, and 177 observations pertaining to care-seeking specifically at hospitals or clinics. The complete analytical datasets can be found in S1 Data.

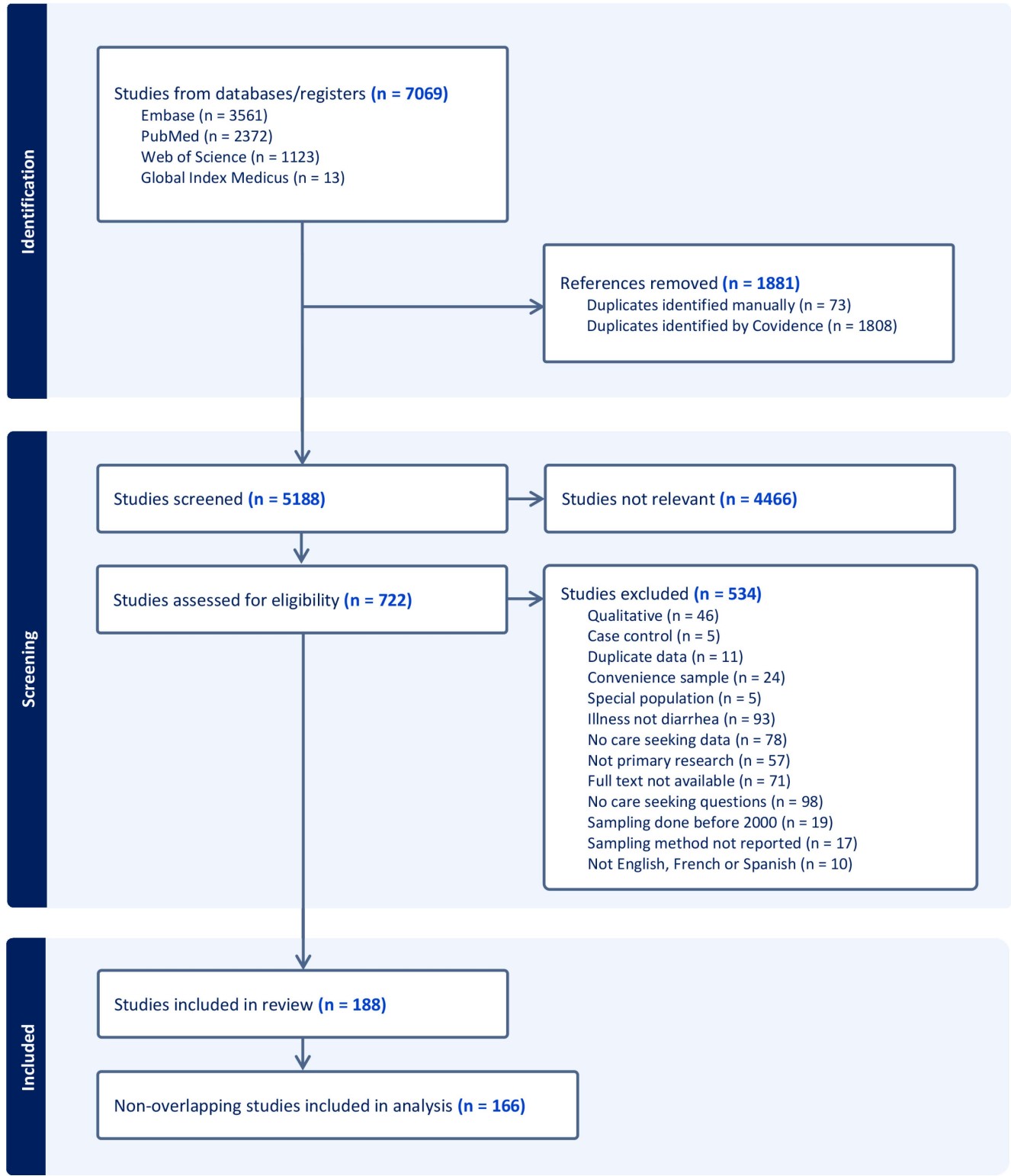

**Fig 1. PRISMA flow diagram.** Diagram illustrates the search, screening, and inclusion process for the systematic review, including databases searched, reasons for exclusion, and final studies included in the meta-analyses.

The studies included in the primary dataset were conducted in 62 countries across 10 World Bank geographic regions, with most observations coming from East Africa (N = 60), South Asia (N = 41), Latin America and the Caribbean (N = 25), Western Africa (N = 24), and East Asia and the Pacific (N = 23). Twenty of the observations came from nationally representative studies, and 53, 78, and 60 came from studies representative of first, second, and third administrative divisions, respectively (Fig A in S1 Appendix). Studies included sampling completion dates between 2000 and 2022, with most of the observations completed during 2005–2009 (N = 67) and 2010–2014 (N = 67) (Fig B in S1 Appendix).

Most observations in the primary dataset were from studies that asked a caregiver about care-seeking for a child with diarrhea (N = 138, 65.4%) (Table 1). Others represented a combination of children, other family members, or the respondent themselves (N = 59, 28.0%) and a small proportion represented care-seeking exclusively for the survey respondents (N = 14, 6.6%) (Table 1). A small proportion of observations represented clinic-base surveillance, where individuals or caregivers were recruited and surveyed at a hospital or clinic about their care-seeking behavior (N = 13, 6.2%) (Table 1). More of the observations were from exclusively rural areas (N = 83, 39.3%) than exclusively urban areas (N = 68, 32.2%), and two (0.9%) came from internally displaced persons or refugees (Table 1). Seventeen (8.1%) observations took place during or closely following a diarrheal disease outbreak (Table 1).

The majority of the observations came from surveys where respondents were asked about care-seeking for general diarrhea (N = 156, 73.8%), followed by either gastroenteritis or non-cholera etiologies (N = 33, 15.6%), and severe diarrhea or cholera including deaths (N = 22, 10.4%) (Table 1). Most observations focused on care-seeking at any time during illness (N = 175, 82.9%) and most used recall periods of 14–30 days (N = 143, 67.8%) (Table 1).

## Trends in care-seeking at hospitals or clinics in the unadjusted data

Our primary focus was care-seeking at hospitals or clinics, where people can receive treatment and are most often counted in disease surveillance efforts [22,23], in low- and middle-income countries (LMICs). Since care-seeking behavior among individuals surveyed at a hospital or clinic likely differs from that of individuals surveyed in the broader community, we excluded observations with clinic-based surveillance from all the following analyses unless otherwise noted.

We found that the reported proportion seeking care at a hospital or clinic in 145 observations from LMICs was higher for observations from studies that surveyed participants about care-seeking at any time during illness (median of 34.7%; N = 125; IQR, 18.9% to 55.0%) compared to their first source of care (median of 21.2%; N = 20; IQR, 10.0% to 33.5%) (Fig 2a) as well as studies conducted during or following a diarrheal disease outbreak (median of 62.7%; N = 11; Interquartile Range (IQR), 44.2% to 65.5%) compared to those that were not (median of 31.0%; N = 134; IQR, 15.3% to 49.9%) (Fig 2c). Care-seeking at hospitals or clinics was also higher for severe diarrhea or cholera (median of 64.1%; N = 15; IQR, 45.4% to 70.0%) compared to general diarrhea (median of 31.0%; N = 122; IQR, 15.3% to 48.2%) and gastroenteritis or other etiologies (median of 33.3%; N = 8; IQR, 25.5% to 44.1%) (Fig 2b). Although we found no significant correlation between overall study quality and reported care-seeking at hospitals or clinics (Spearman rho = 0.07; 95% Confidence Interval -0.09 to 0.23; Fig E in S1 Appendix), studies with internal inconsistencies identified during study quality assessment more often reported lower care-seeking than those without (Fig E in S1 Appendix).

Studies that surveyed participants about their own care-seeking behavior reported lower levels of care-seeking on average compared to those focused on children (Fig 2d). Yet, across 134 observations in LMICs that reported the age distribution of participants, we found no significant correlation between the proportion of the study population under five years of age

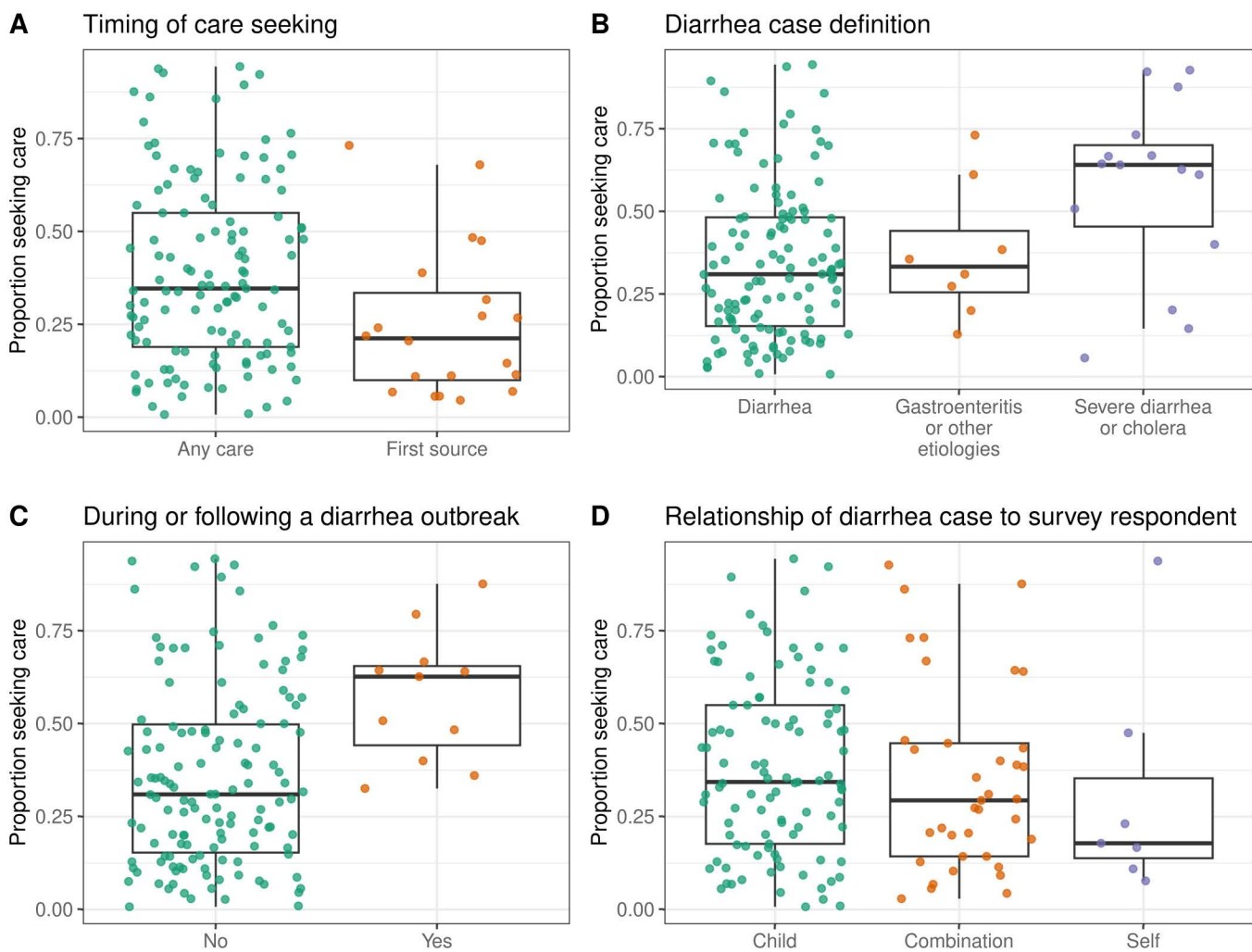

**Fig 2. Proportion that sought care at a hospital or clinic in LMICs by observation characteristics.** Proportion of respondents that reported seeking care for diarrheal illness at a hospital or clinic grouped by **A)** whether the study surveyed participants about care-seeking at any time during illness or their first source of care, **B)** survey respondent category, **C)** whether or not the study described an outbreak, and **D)** whether the care-seeking was for the respondent themselves, their child, or other, which often included any household member. Each point represents an observation (N = 145). Boxes represent the median and interquartile range (IQR) of the proportion for each group. Lines extend from the top and bottom of box to the largest proportion value no further than 1.5 * IQR from the box.

and the proportion that sought care at hospitals or clinics (Spearman rho = 0.10; 95% Confidence Interval -0.08 to 0.26; Fig 3a). Studies conducted only among children reported higher care-seeking at hospitals or clinics (median of 34.2%; N = 98; IQR, 17.5% to 55.0%) compared to studies conducted only among adults or individuals over five (median of 22.4%; N = 22; IQR, 14.5% to 49.3%) (Fig 3c), but there was wide variation between studies and this trend held for diarrhea and gastroenteritis but not for severe diarrhea or cholera (Fig 3b, d).

## Trends in care-seeking at other sources outside the home in LMICs

Hospitals and clinics were the most common source of care sought across studies in LMICs (median of 32.3%; N = 145; IQR, 17.0% to 52.6%) (Fig 4a), though they were less commonly the first source of care (median of 21.2%; N = 20; IQR, 10.0% to 33.5%) (Fig 4b). Seeking care from community health workers (CHW) was reported more often as first sources of care (median of

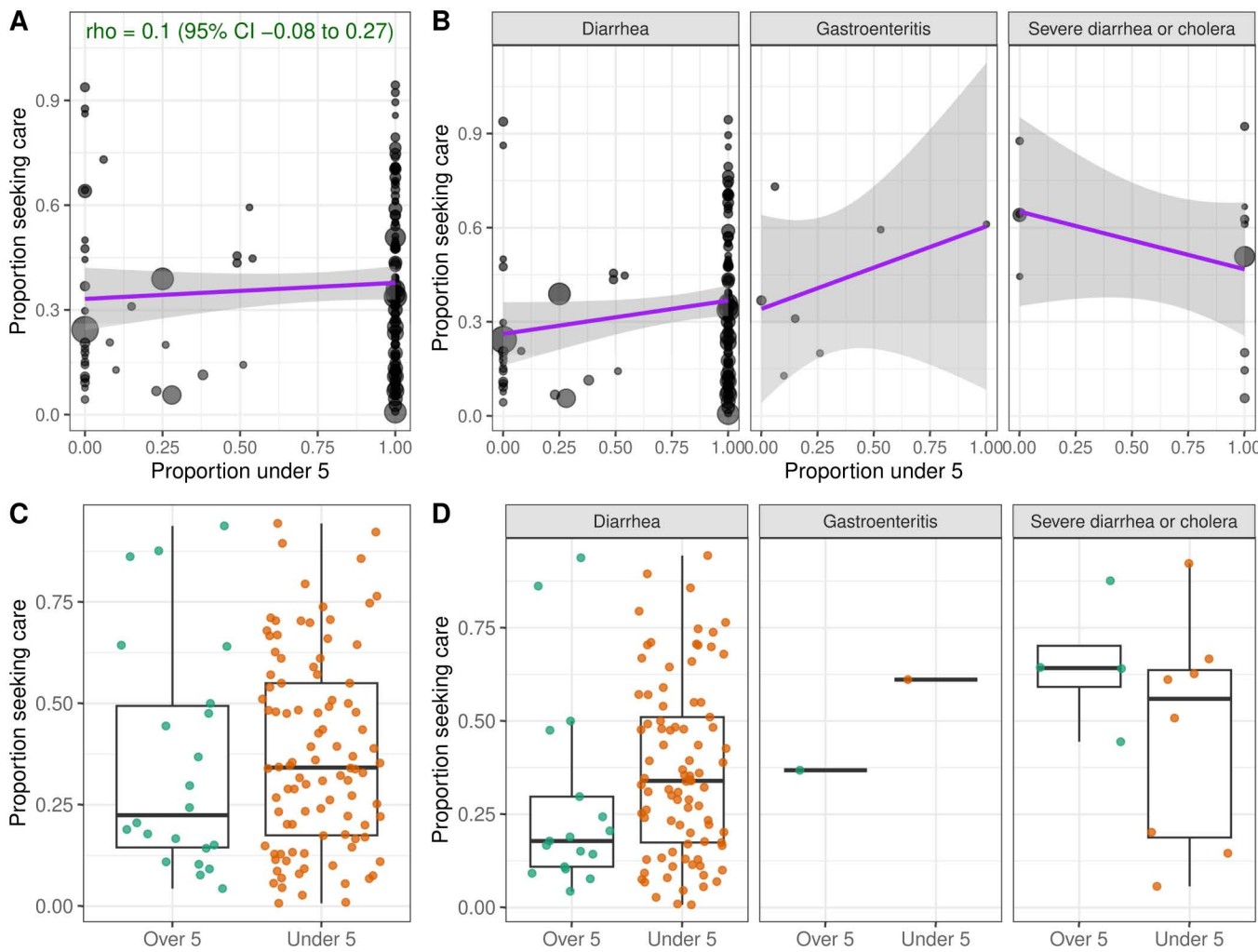

**Fig 3. Relationship between age and care-seeking at hospitals or clinics in LMICs. A-B)** Relationship between the proportion of respondents that reported seeking care for diarrheal illness at a hospital or clinic (y-axis) and the proportion of the study population under five years of age (x-axis), **A)** overall and **B)** stratified by diarrhea case definition. Size of points is proportional to the number of survey respondents per observation (N = 134). Spearman's rho is shown with 95% confidence interval (CI) in green. Linear relationships are shown with purple lines. **C-D)** Proportion seeking care grouped by studies that were exclusively among children under five (N = 98) or exclusively among individuals over five (N = 22), **C)** overall and **D)** stratified by diarrhea case definition.

16.9%; N = 9; IQR, 4.5% to 18.9%) than the main or primary of care (median of 6.2%; N = 32; IQR, 1.7% to 14.7%) (Fig 4b). Seeking care from CHW and health posts were also more frequently reported in observations from low-income compared to middle-income countries (Fig F in S1 Appendix) and for general diarrhea compared to severe diarrhea or cholera (Fig G in S1 Appendix). Traditional healers were rarely reported as a source of care (median of 4.0%; N = 48; IQR, 1.0% to 8.7%) (Fig 4). These less common sources were also less frequently included or reported as options in survey questions than hospitals or clinics (Fig 4).

## Trends in care-seeking for clinic-based surveillance studies in LMICs

In clinic-based surveillance studies that surveyed participants at a hospital or clinic, which were excluded in all other analyses, most participants reported hospitals/clinics as their first source of care (median of 89.3%; N = 4; IQR, 74.8% to 97.7%) (Fig H in S1 Appendix). In

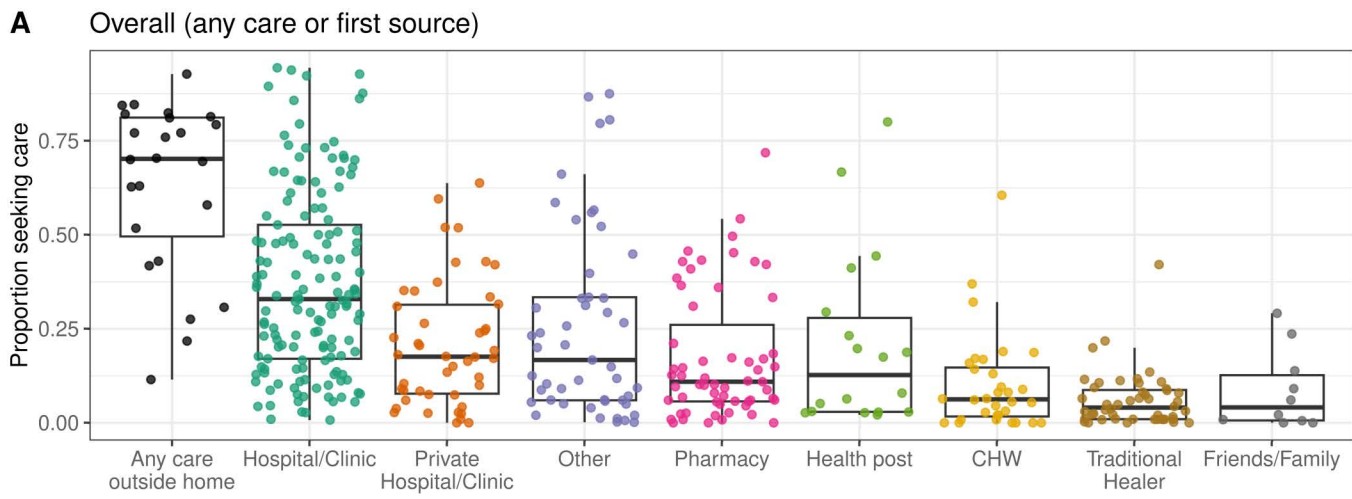

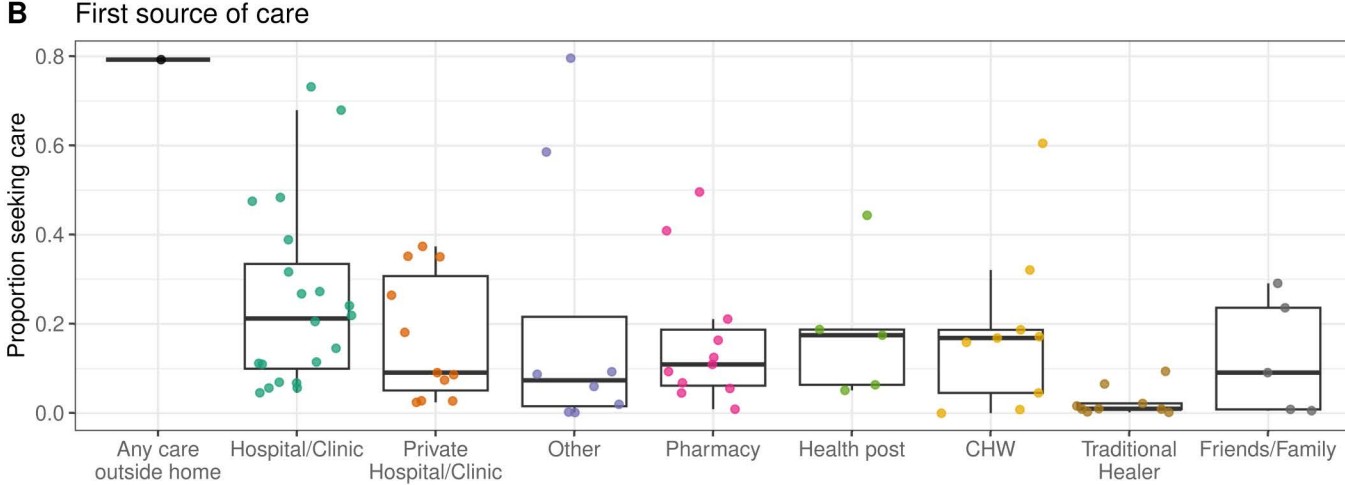

**Fig 4. Proportion that sought care by source in LMICs.** Proportion of respondents that reported seeking care for diarrheal illness at all categories of care sources, including any care outside of the home, hospitals or clinics including public facilities, private hospitals or clinics, pharmacies or kiosks, health posts, community healthcare workers (CHW), traditional healers, and friends or family, for **A)** all studies and **B)** the subset of studies that asked survey respondents about the first source of care they sought. Each point represents an observation.

studies that surveyed participants about where they sought care before their hospital or clinic visit, hospitals/clinics were the most common previous sources of care, (median of 35.1%; N = 2; IQR, 31.2% to 39.0%), followed by pharmacies (median of 20.7%; N = 2; IQR, 18.7% to 20.4%), private hospitals/clinics (median of 10.6%; N = 2; IQR, 9.4% to 11.9%), and traditional healers (median of 5.6%; N = 2; IQR, 5.4% to 5.8%) (Fig H in S1 Appendix).

## Adjusted estimates of care-seeking at hospitals or clinics in LMICs

Using a random-effects meta-analysis, we estimated that on average 32.8% (95% Confidence Interval (CI) 28.1% to 37.9%) of individuals sought care at hospitals or clinics across 145 observations in LMICs (Fig I in S1 Appendix). In addition, we predicted that care-seeking for a future study would be in the range of 3.3% to 87.5% (Fig I in S1 Appendix). This wide prediction interval reflects the substantial variation we found between studies in care-seeking ($\tau^2$ = 1.8; $I^2$ = 99.3; 95% CI, 99.2 to 99.3), with adjusted study-level estimates of care-seeking

ranging from 0.7% (95% CI 0.5% to 0.9%) to 94.4% (95% CI 90.6% to 97.0%) (Fig I in S1 Appendix).

We estimated that the proportion of people that would seek care during a hypothetical diarrheal episode, as opposed to an actual previous episode, in LMICs was substantially higher, with 71.2% (95% CI 57.1% to 83.2%; prediction interval 5.8% to 99.1%) intending to seek care at a hospital or clinic if they or their child had diarrhea across 29 observations (Fig J in S1 Appendix).

### Factors associated with variation in care-seeking at hospitals or clinics in LMICs

We next examined factors associated with variation in care-seeking in LMICs. In univariate analyses, we found that the odds of seeking health care at a hospital or clinic were higher for severe diarrhea or cholera than general diarrhea, as well as during or after an outbreak and when a recall period exceeded 30 days (Table B in S1 Appendix). If the question was restricted to the first source of care sought, the odds of seeking care at a hospital or clinic were lower (Table B in S1 Appendix). We found no significant differences in care-seeking between observations conducted among only children under five compared to those conducted only among individuals over five (OR 1.06 (95% CI 0.55 to 2.01)) (Table 2). In a subset of eight studies with results stratified by age, differences in care-seeking for individuals under five years of age compared to over five years remained non-significant, though there was a trend towards higher care-seeking for children under five (OR 1.24 (95% CI 0.29 to 5.36)) (Table 2).

To examine potential confounding between the sources of variation in our dataset, we additionally tested for associations between the variables related to study design and setting specific attributes included in the univariate analyses above. We found that studies with severe diarrhea or cholera case definitions were associated with longer recall periods and were more likely to have been conducted during or after an outbreak compared with studies on general diarrhea (Table C in S1 Appendix). In sub-analysis, we found that there was no longer an effect of being in an outbreak setting when subsetting to studies with severe diarrhea or cholera case definitions (Table D in S1 Appendix). In addition, there was no longer an effect of recall period when subsetting the data to observations with general diarrhea case definitions (Table D in S1 Appendix).

Since we could not separate the effects of case definition from recall period or outbreak context, we examined the effects of case definition on care-seeking at hospitals or clinics adjusting only for the timing of care-seeking (i.e., whether they were surveyed about their first source of care or care at any time during illness). The adjusted odds of seeking care for severe diarrhea or cholera remained significantly higher than for general diarrhea (OR 3.43 (95% CI 1.71 to 6.88)) (Table 3), and residual heterogeneity remained high ($\square^2 = 1.57$; $I^2 = 99.5$; $R^2 = 10.2\%$).

**Table 2. Sub-analyses of age and care-seeking at hospitals or clinics in LMICs.** Odds that an individual seeks care for diarrheal illness for a child under five years of age compared to an adult or individual over five years of age. Results from two sub-analyses are shown: 1) studies only among children or only among adults, and 2) studies with care-seeking stratified for children and adults. p-values are shown for the univariate mixed-effects models; n.s. indicates not significant (p > 0.05).

| Sub-analysis | Age category | Odds ratio | p-value |
|---|---|---|---|
| Observations among only children (N = 98) or only adults (N = 22) | Over five | 1 [Reference] | |
| | Under five | 1.06 (0.55 - 2.01) | n.s. |
| Observations with care-seeking stratified by age (n = 8) | Over five | 1 [Reference] | |
| | Under five | 1.24 (0.29 - 5.36) | n.s. |

**Table 3. Factors associated with variation in care-seeking at hospitals or clinics in LMICs. Odds that an individual sought care for diarrheal illness for severe diarrhea or cholera and gastroenteritis or other etiologies compared to general diarrhea, adjusting for timing of care-seeking. p-values are shown for the multivariate mixed-effects model; n.s. indicates not significant (p > 0.05).**

| Variable | Category | Odds ratio | p-value |
|---|---|---|---|
| Diarrhea case definition | Diarrhea | 1 [Reference] | |
| | Severe diarrhea or cholera | 3.43 (1.71 - 6.88) | < 0.001 |
| | Gastroenteritis or other etiologies | 1.15 (0.46 - 2.90) | n.s. |
| Timing of care-seeking | Care sought at any time | 1 [Reference] | |
| | First source sought | 0.44 (0.24 - 0.80) | < 0.01 |

Correspondingly, when we stratified the meta-analysis by case definition, we estimated that 58.6% (95% CI 39.9% to 75.2%; Prediction Interval 5.2% to 97.3%) of individuals sought care at a hospital or clinic when they or their child had severe diarrhea or cholera (Fig M in S1 Appendix) compared to 29.9% (95% CI 25.3% to 35.1%; Prediction Interval 3.1% to 84.9%) for general diarrhea (Fig L in S1 Appendix) and 35.8% (95% CI 23.7% to 50.0%; Prediction Interval 6.5% to 81.8%) for gastroenteritis and non-cholera etiologies (Fig N in S1 Appendix).

## Discussion

Our findings reveal that most people with diarrhea do not seek care at hospitals or clinics. This is important as frequently only individuals attending hospital are counted in typical disease surveillance systems. Furthermore, it may indicate gaps in treatment access. We also found substantial variation between studies, even after accounting for sources of variation such as case definition and timing of care-seeking. Hospitals and clinics were the most common source of care investigated and reported, with pharmacies, CHW, and health posts less commonly reported, and traditional healers very rarely cited as a source of care. Studies that phrased questions in terms of hypothetical care-seeking reported substantially higher care-seeking than those that asked about actual diarrheal episodes. We found no significant differences in care-seeking between children under five and older individuals, and the odds of seeking care for severe diarrhea and cholera were significantly higher compared to general diarrhea.

Our findings shed light on the extent to which we may underestimate diarrhea burden when we rely solely on passive surveillance data, with the greatest underestimates occurring, unsurprisingly, among individuals with mild or moderate symptoms. Additionally, our findings suggest that the relationship between age and care-seeking behavior varies by location, context, and etiology. While we found trends towards adults seeking care less often for themselves than for children with diarrhea, we found that, on average, adults sought care for severe diarrhea and cholera at similar rates as for children. Given these results and the considerable heterogeneity across studies, we recommend that local data on care-seeking behavior, stratified by age and either symptom severity or etiology, be used to adjust burden estimates whenever possible. In the absence of local data or for global analyses, the case-definition-stratified estimates from this review may provide useful approximations of missed symptomatic cases.

Although we estimated lower average care-seeking rates than previous analyses of DHS data (33% here vs. 45–51% in DHS data)—possibly due to our inclusion of a broader range of study settings and populations—our findings of very low reported care-seeking at CHW and traditional healers align with those studies [5–9]. This may reflect true patterns of care-seeking behavior, but it could also indicate a research bias, as hospitals and clinics were more frequently included in survey questionnaires than other sources of care. Additionally, only a minority of studies reported results where multiple sources of care were selected, and

individuals may be less likely to report CHW or traditional healers as their sole or primary source of care. This is supported by our finding that CHW were reported more frequently, and hospitals and clinics less frequently, in studies that examined the first source of care rather than overall source of care. These limitations could have led us and others to under-estimate the role of CHW and traditional healers in diarrhea case management.

This study has several additional limitations. Only a small proportion of the included studies were conducted among adults or individuals over five years old and care-seeking in those studies varied widely, which may have prevented us from identifying significant trends. Methodology and questionnaires also varied between studies, which we may not have fully captured in our analysis, and there were several variables we could not examine due to data scarcity or potential confounding (e.g., income group, outbreak setting, and recall period).

We also estimated that more than twice as many people would seek care as actually did, but we were not able to examine factors that shape care-seeking behavior in detail. Decisions about whether and where to seek care are complex, with tradeoffs and determining factors that vary by location, time, and individuals. Existing theories for health service utilization describe a wide range of factors influencing care-seeking, including environmental and health system factors (e.g., provider and treatment availability, transportation and access), population and socio-demographic factors (e.g., age, gender, income, culture, community support, disease prevalence), and individual behavior (e.g., perceptions about disease risk and severity, health care costs, and benefits) [26]. The huge heterogeneity we found in care-seeking, even after accounting for different aspects of study methodology, may result from the propagation of individual heterogeneity in decision making and contextual factors. An important area for future research will be to examine these factors to identify ways in which care-seeking could be improved, especially for severe cases.

Given our findings, and their limitations, we propose several areas for future work. First, it will be important for future studies to include questions about care-seeking across ages and diarrhea case definitions, include a broad range of sources of care including health posts and CHW, and focus on actual rather than hypothetical care-seeking where possible. In addition, studies that explicitly examine the order that individuals seek care at different sources (for example, [27–29]) can help inform when and why there may be delays in seeking care at hospitals or clinics and how this varies by age and disease severity. Finally, synthesizing qualitative studies that examine reasons why individuals do or do not seek care could provide insight into non-hospital-based care-seeking as well as what needs to be done to reduce barriers for populations with the least interaction with the health system.

Our findings suggest that care-seeking estimates for broadly defined diarrhea may not reflect true patterns, particularly for severe cases. While we observed trends across age groups, differences in care-seeking between children under five and older individuals were not statistically significant, underscoring the need for more granular data. DHS data can provide a reasonable approximation of care-seeking for non-severe, non-etiology-specific diarrhea, but assumptions about age-related care-seeking should be setting-specific, as adult behaviors varied across contexts. More accurate estimates of care-seeking behaviors can help translate imperfect observations of disease in the population into more reliable assessments of true burden, ultimately supporting the design of better interventions to ensure timely access to appropriate diarrhea treatment across different populations.

## Supporting information

**S1 Checklist.  Preferred Reporting Items for Systematic Reviews and Meta-Analyses (PRISMA) 2020 Checklist.**
(PDF)

**S1 Appendix. Supporting information.** Supplementary methods, including systematic review search terms, and supplementary figures and tables.
(PDF)

**S1 Data. Full dataset.** Excel sheet with the complete data extracted from all studies that met the inclusion criteria (tab 1) as well as all variable descriptions (tab 2). Data extracted from the nonoverlapping studies included in the main analysis dataset can be found by filtering for the values "1" in the column "Primary dataset.".
(XLSX)

**S2 Data. Studies that were assessed for eligibility.** Excel sheet with citation information for all 722 studies that underwent full-text screening, including citation details for 534 studies that were excluded along with the reason for exclusion (tab 1) and citation details for 188 studies that met the inclusion criteria (tab 2).
(XLSX)

## Author contributions

**Conceptualization:** Kirsten E Wiens, Elizabeth C. Lee, Andrew S. Azman.

**Data curation:** Marissa H. Miller, Daniel J. Costello, Ashlynn P. Solomon.

**Formal analysis:** Kirsten E Wiens.

**Funding acquisition:** Kirsten E Wiens, Andrew S. Azman.

**Methodology:** Kirsten E Wiens, Andrea G. Shipper, Elizabeth C. Lee, Andrew S. Azman.

**Project administration:** Kirsten E Wiens.

**Validation:** Kirsten E Wiens, Marissa H. Miller, Skye M. Hilbert.

**Visualization:** Kirsten E Wiens.

**Writing – original draft:** Kirsten E Wiens, Andrea G. Shipper.

**Writing – review & editing:** Kirsten E Wiens, Marissa H. Miller, Daniel J. Costello, Ashlynn P. Solomon, Skye M. Hilbert, Andrea G. Shipper, Elizabeth C. Lee, Andrew S. Azman.

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
