## [Decision Letter · Decision Letter 0]

24 Jan 2025

PGPH-D-24-02929

Care seeking for diarrheal illness: a systematic review and meta-analysis

Dear Dr. Wiens,

Thank you for submitting your manuscript to PLOS Global Public Health. We would like to invite you to submit a revised version of the manuscript based on the inputs and suggestions by the reviewers who have carefully looked into the systematic review

We look forward to receiving your revised manuscript.

Kind regards,

Sindhu Kulandaipalayam Natarajan, MBBS, MD, DTM&H, PhD

Academic Editor

Journal Requirements:

2. Please provide an Author Summary. This should appear in your manuscript between the Abstract (if applicable) and the Introduction, and should be 150–200 words long. The aim should be to make your findings accessible to a wide audience that includes both scientists and non-scientists. Sample summaries can be found on our website under Submission Guidelines:

https://journals.plos.org/globalpublichealth/s/submission-guidelines#loc-parts-of-a-submission.

3. As required by our policy on Data Availability, please ensure your manuscript or supplementary information includes the following: 

**Comments to the Author**

1. Does this manuscript meet PLOS Global Public Health’s publication criteria ? Is the manuscript technically sound, and do the data support the conclusions? The manuscript must describe methodologically and ethically rigorous research with conclusions that are appropriately drawn based on the data presented.

Reviewer #1: Partly

Reviewer #2: Yes

2. Has the statistical analysis been performed appropriately and rigorously?

Reviewer #1: Yes

Reviewer #2: Yes

3. Have the authors made all data underlying the findings in their manuscript fully available (please refer to the Data Availability Statement at the start of the manuscript PDF file)?

Reviewer #1: Yes

Reviewer #2: Yes

4. Is the manuscript presented in an intelligible fashion and written in standard English?

Reviewer #1: Yes

Reviewer #2: Yes

5. Review Comments to the Author

Reviewer #1: Overall: This is a systematic review and meta-analysis focusing on global care-seeking behaviors for diarrheal illness. The review and data abstraction was methodically sound. However, due to the large volume of data abstracted and analyzed, there are some sections where the data stratifications (i.e., which data are included in a particular analysis) are unclear. In addition, the broad scope of the paper and number of results makes it difficult to draw meaningful conclusions. It would be helpful to narrow down the scope of the paper (e.g., care-seeking behaviors for hospitals and clinics in LMICs) and limit conclusions to those that directly support the research questions.

Specific comments:

• Lines 186 – 188: Are there any citations to support the statement that private hospitals are rarely part of surveillance systems? This likely varies greatly by country, and if these are truly not generally part of surveillance systems it would underscore a need to capture care-seeking and diarrheal burden in these facilities. If separating these out, it would be worth conducting a sensitivity analysis comparing care-seeking behaviors among those explicitly defined as private hospitals with public hospitals.

• Lines 258 – 262: The authors state that they have excluded clinics from this section unless otherwise stated, but most of the section (i.e., lines 272 – 285) seems to include clinics. For clarity, would suggest completely separating these studies out into their own section, or including them and conducting a sensitivity analysis.

• Lines 413 – 415: If the survey questions/underlying data represent where respondents would seek care, it may not be an appropriate conclusion to state that a certain percentage would not seek care at hospitals/clinics.

• Please describe any potential differences in care-seeking behaviors in papers that included sampling during the COVID-19 pandemic.

Reviewer #2: This is an excellent piece of research and deserves to be published with only some minor stylistic revisions. The methodological rigor of the review is excellent, with methods described in wonderful detail, and with transparency throughout. The paper addresses an important research question which is clearly outlined, and the authors illustrate the gap their research fills. The only revisions I would recommend are minor linguistic/stylistic edits as follows:

I would argue that the first sentence of the abstract would read better as "Monitoring and treating diarrheal illness often relies...", instead of "Monitoring and treating diarrheal illness rely.." as I feel the subject of the sentence "Monitoring and treating diarrheal illness" is essentially singular here not plural, as it acts as an unified singular concept, grammatically being a compound gerund phrase.

Otherwise, I would ensure that "care seeking" is written as "care-seeking" throughout the text. Both versions are currently used, with "care seeking" featuring throughout the piece.

I feel the first line of the discussion could be stronger. "Here we estimated that most people with diarrhea (on average 67%) do not seek care at hospitals or clinics where they can receive treatment when needed and be counted in typical surveillance systems". Perhaps something like "Our findings reveal that most people with diarrhea do not seek care at hospitals or clinics. This is important as frequently only individuals attending hospital are counted in typical disease surveillance systems. Furthermore, it may indicate gaps in treatment access".

I also think lines 193-195 could be improved. It could be worthwhile explicitly stating why the maximum values were taken, given the risk of overestimation. However, I understand this method is simple and consistent. The phrasing itself is quite complicated: "When aggregating data from studies with multiple choice questions where there were multiple different options within one category or source of care (e.g., multiple different hospitals or clinics that could have been selected), we took the maximum value across those options." Something along the lines of the following sentence may be clearer:

"For studies with multiple-choice questions that provided several options within the same category (e.g., public hospitals, private hospitals, or community clinics under 'source of care'), we used the highest percentage reported within the category to represent the overall care-seeking behavior for that category in our analysis."

Lastly, I would revise the final paragraph of the article. The conclusion feels a little weak and I feel could be improved to highlight the importance of your piece and its findings, the urgency to address gaps and how your findings can drive actionable change.

Overall though, this is a great piece of work and deserves to be published.

6. PLOS authors have the option to publish the peer review history of their article (what does this mean? ). If published, this will include your full peer review and any attached files.

**Do you want your identity to be public for this peer review?** For information about this choice, including consent withdrawal, please see our Privacy Policy .

Reviewer #1: No

Reviewer #2: No

---

## [Decision Letter · Decision Letter 1]

20 Mar 2025

Care-seeking for diarrheal illness: a systematic review and meta-analysis

PGPH-D-24-02929R1

Dear Dr Wiens,

We are pleased to inform you that your manuscript 'Care-seeking for diarrheal illness: a systematic review and meta-analysis' has been provisionally accepted for publication in PLOS Global Public Health.

Best regards,

Sindhu Kulandaipalayam Natarajan, MBBS, MD, DTM&H, PhD

Academic Editor

**Comments to the Author**

1. If the authors have adequately addressed your comments raised in a previous round of review and you feel that this manuscript is now acceptable for publication, you may indicate that here to bypass the “Comments to the Author” section, enter your conflict of interest statement in the “Confidential to Editor” section, and submit your "Accept" recommendation.

Reviewer #2: All comments have been addressed

2. Does this manuscript meet PLOS Global Public Health’s publication criteria ? Is the manuscript technically sound, and do the data support the conclusions? The manuscript must describe methodologically and ethically rigorous research with conclusions that are appropriately drawn based on the data presented.

Reviewer #2: Yes

3. Has the statistical analysis been performed appropriately and rigorously?

Reviewer #2: Yes

4. Have the authors made all data underlying the findings in their manuscript fully available (please refer to the Data Availability Statement at the start of the manuscript PDF file)?

Reviewer #2: Yes

5. Is the manuscript presented in an intelligible fashion and written in standard English?

Reviewer #2: Yes

6. Review Comments to the Author

Reviewer #2: All comments addressed appropriately. The paper reads very well now! Great work! This is a meaningful contribution to the literature.

7. PLOS authors have the option to publish the peer review history of their article (what does this mean? ). If published, this will include your full peer review and any attached files.

**Do you want your identity to be public for this peer review?** For information about this choice, including consent withdrawal, please see our Privacy Policy .

Reviewer #2: No
